# Angiotensin II Exaggerates SARS-CoV-2 Specific T-Cell Response in Convalescent Individuals following COVID-19

**DOI:** 10.3390/ijms23158669

**Published:** 2022-08-04

**Authors:** Moudhi Almutlaq, Fatmah A. Mansour, Jahad Alghamdi, Yassen Alhendi, Abir Abdullah Alamro, Amani Ahmed Alghamdi, Hassan S. Alamri, Fayhan Alroqi, Tlili Barhoumi

**Affiliations:** 1Biochemistry Department, College of Science, King Saud University, Riyadh 11451, Saudi Arabia; 2Medical Research Core Facility and Platforms, King Abdullah International Medical Research Center, King Abdulaziz Medical City, Ministry of National Guard Health Affairs, Riyadh 11426, Saudi Arabia; 3Saudi Biobank, King Abdullah International Medical Research Center, King Saud Bin Abdulaziz University for Health Sciences, Ministry of National Guard Health Affairs, Riyadh 11426, Saudi Arabia; 4Clinical Laboratory Sciences, King Saud Bin Abdulaziz University for Health Sciences, Riyadh 11426, Saudi Arabia; 5Department of Pediatrics, King Abdulaziz Medical City, King Abdullah Specialized Children’s Hospital, Riyadh 14611, Saudi Arabia

**Keywords:** renin−angiotensin system, angiotensin II, COVID-19, hypertension, inflammation

## Abstract

Dysregulation of renin−angiotensin systems during coronavirus disease 2019 (COVID-19) infection worsens the symptoms and contributes to COVID-19 severity and mortality. This study sought to investigate the effect of exogenous angiotensin II (Ang-II) on severe acute respiratory syndrome coronavirus 2 (SARS-CoV-2)-specific T-cells response in recovered COVID-19 patients. Human peripheral blood mononuclear cells (PBMCs) were treated with Ang II and then stimulated with a SARS-CoV-2 peptide pool. T-cell responses were measured using flow cytometry, while enzyme-linked immunosorbent assay (ELISA) and intracellular cytokine staining (ICS) assays determined functional capability and polarization. Additionally, the relative level of protein phosphorylation was measured using a phosphokinase array. Our results showed that Ang II treatment significantly increased the magnitude of SARS-CoV-2-specific T-cell response in stimulated PBMCs with a SARS-CoV-2 peptide pool. Moreover, the phosphorylation levels of numerous proteins implicated in cardiovascular diseases, inflammation, and viral infection showed significant increases in the presence of Ang II. The mitogenic stimulation of PBMCs after Ang II and SARS-CoV-2 peptide pool stimulation showed functional polarization of T-cells toward Th1/Th17 and Th17 phenotypes, respectively. Meanwhile, ELISA showed increased productions of IL-1β and IL-6 in Ang II-stimulated PBMCs without affecting the IL-10 level. To our knowledge, this study is the first to demonstrate that Ang II exaggerates SARS-CoV-2-specific T-cells response. Therefore, during COVID-19 infection, Ang II may aggravate the inflammatory response and change the immune response toward a more inflammatory profile against SARS-CoV-2 infection.

## 1. Introduction

Coronavirus disease 2019 (COVID-19) is an emerging respiratory disease caused by severe acute respiratory syndrome coronavirus 2 (SARS-CoV-2) [1]. The presence of underlying comorbidities represents one of the primary contributors to the severity of COVID-19. Among 1786 COVID-19 patients, hypertension (15.8%), cardiovascular diseases (11.7%), and diabetes (9.4%) were the most prevalent comorbidities in severe cases [2]. The angiotensin-converting enzyme 2 (ACE-2), SARS-CoV-2 entry portal, is known to regulate blood pressure through the conversion of angiotensin I (Ang I), the angiotensin II (Ang II) precursor, to angiotensins 1–9 to reduce Ang II production [3] or the conversion of Ang II to angiotensins 1–7 [4]. The binding of SARS-CoV-2 with the ACE-2 receptor during COVID-19 infection was suggested to reduce ACE-2 levels and subsequently increase Ang II accumulation, leading to classical renin angiotensin system (RAS) dysregulation. Osman et al. found that ACE-2 mRNA expression and plasma levels decreased, while Ang I and Ang II levels increased in COVID-19 patients compared to in the healthy control [5]. Another study conducted by Liu et al. showed an increased Ang II level in COVID-19 patients that bore a positive correlation with viral load and lung injury [6].

Therefore, it seems that the harmful consequences related to COVID-19, such as hypercoagulation, endothelial dysfunction, and inflammation, stem from the presence of previous RAS dysregulation as explained in our previous study [7]. In the presence of SARS-CoV-2 infection, dysregulated RAS as in hypertension, atherosclerosis, and diabetes worsens the symptoms and contributes to COVID-19 severity and mortality [8]. To address such a knowledge gap, this study assessed the effect of exogenous Ang II on SARS-CoV-2-specific T-cells response in peripheral blood mononuclear cells (PBMCs) obtained from recovered COVID-19 patients and stimulated with a SARS-CoV-2 peptide pool. 

## 2. Results

### 2.1. Ang II Exaggerates SARS-CoV-2-Specific Memory T-cell Response in Recovered COVID-19 Patients

The activation-induced cell marker (AIM) assay determined the effect of Ang II on SARS-CoV-2 peptide pool-stimulated PBMCs from recovered patients. Stimulation with a SARS-CoV-2 peptide pool alone prompted an increase of CD8+ (CD38+, CD69+, CD127+, and CD154+) T-cell activation markers (Figure 1 and Figure 2A) and CD4+ (CD38+, CD127+, and CD154+, but not CD69+) T-cell activation markers (Figure 2B and Figure 3) relative to those in the unstimulated control group, indicating SARS-CoV-2-specific memory T-cell response. The stimulation of PBMCs with 100 nM Ang II for 24 h followed by stimulation with a SARS-CoV-2 peptide pool resulted in a significant exaggeration of SARS-CoV-2-specific memory T-cell response. Such a response was indicated by an increase of both CD4+ (CD38+ and CD69+) and CD8+ and CD69+ T-cells relative to those stimulated with a SARS-CoV-2 peptide pool alone (*p* < 0.05; Figure 1B and Figure 3A,B). Additionally, there was a trend of exaggeration in CD4+ CD127+ and CD8+ (CD38+ and CD127+) responses as shown in (Figure 1A,C and Figure 3C). By contrast, CD4+ CD154+ and CD8+ CD154+ responses were dampened in the presence of both Ang II and the SARS-CoV-2 peptide pool relative to in the presence of the SARS-CoV-2 peptide pool alone (Figure 2). During inflammation, cardiac fibrosis and hypertension CD4+ CD25+ regulatory T-cells activation proved integral in suppressing inflammatory signals and ameliorating organ damage [9,10,11]. In this study, CD4+ CD25+ and CD8+ CD25+ activation markers did not show any differential responses among the stimulated groups compared with those in the control group (Appendix A). Therefore, the presence of Ang II appeared to promote the activation of inflammatory response markers without affecting anti-inflammatory ones.

### 2.2. Ang II Alters the Inflammatory Response in Recovered COVID-19 Patients

IFN-γ and TNF-α together synergize the killing of infectious agents during bacterial [12] and parasitic diseases [13]. In this study, the response to the SARS-CoV-2 peptide pool and Ang II stimulation separately indicated an increase of TNF-α and IFN-γ-expressing CD4+ T-cells and TNF-α, but not IFN-γ-expressing CD8+ T-cells relative to in the unstimulated control group (Figure 4 and Appendix A). In pre-stimulated PBMCs with Ang II followed by stimulation with a SARS-CoV-2 peptide pool, CD4+ and CD8+ T-cells responses were altered by decreasing TNF-α expression in both types of T-cell and increasing IFN-γ expression in the CD4+ subset relative to those in the SARS-CoV-2 peptide pool-stimulated group (Figure 4 and Appendix A). This result indicated the possible role of Ang II and hence RAS in altering the inflammatory response. 

### 2.3. Ang II Stimulates T-Helper Type1 (Th1) and Type17 (Th17) Phenotypes in Recovered COVID-19 Patients

Restimulated PBMCs with phorbol 12-myristate 13-acetate (PMA) and ionomycin for 6 h were used to assess the T-cell functional polarization. In the SARS-CoV-2 peptide pool-stimulated group, CD4+ produced IL-17, TNF-α, and IFN-γ (Figure 5A–C), indicating CD4+ polarization towards the Th1/Th17 profile. Furthermore, the stimulation of PBMCs with Ang II followed by the SARS-CoV-2 peptide pool stimulation showed more CD4+ T-cells skewed toward Th1/Th17 phenotype by increasing IL-17, TNF-α, and IFN-γ production compared with those stimulated with SARS-CoV-2 spike glycoprotein alone (Figure 5A–C). Additionally, the CD8+ T-cells were not affected by SARS-CoV-2 stimulation alone (Figure 5D and Appendix A). However, the presence of Ang II stimulation followed by the SARS-CoV-2 peptide pool stimulation showed CD8+ T-cells’ polarization toward the Th17 phenotype by increasing IL-17 production without affecting TNF-α or IFN-γ production (Figure 5D and Appendix A).

### 2.4. Ang II Contributes to the Cytokine Storm Caused by SARS-CoV-2 Infection

The functional capability of the SARS-CoV-2-specific T-cells producing cytokines secreted from stimulated T-cells was measured using ELISA. T-cells produced high levels of proinflammatory IL-1β and low levels of anti-inflammatory IL-10 cytokines (Figure 6A,C). Such findings indicated an inflammatory functional response of memory T-cells against SARS-CoV-2 peptide pool stimulation. Interestingly, pre-stimulated T-cells with Ang II promoted more inflammatory response to the SARS-CoV-2 peptide pool stimulation by increasing IL-1β and IL-6 secretions by ~1.5-fold compared with that in the SARS-CoV-2 peptide pool-stimulated group without affecting the reduction of IL-10 caused by SARS-CoV-2 peptide pool stimulation (Figure 6). This observation suggested that RAS dysregulation may contribute to organ damage caused by the cytokine storm through potentiating inflammatory response during COVID-19 infection.

### 2.5. Ang II Increases the Phosphorylation Level of Kinases in Recovered PBMCs

The Human Phospho-Kinase Array was used with Ang II-stimulated PBMCs to screen the activated proteins in response to SARS-CoV-2 peptide pool stimulation. This approach saw a significant increase in four phosphoproteins: cyclic AMP response element-binding protein (CREB), heat shock protein 27 (HSP27), Src family kinases (Src and Lck), and platelet-derived growth factor receptor (PDGFR β) in stimulated PBMCs with Ang II followed by SARS-CoV-2 peptide pool stimulation (*p* < 0.05 versus SARS-CoV-2 peptide pool stimulated group) (Figure 7). Additionally, a considerable increase occurred in other phosphoproteins, such as glycogen synthase kinase 3 alpha (GSK3α), p38 mitogen-activated protein kinases (p38α), endothelial nitric oxide synthase (eNOS), and extracellular signal-regulated kinases1/2 (ERK1/2) in PBMCs stimulated with Ang II followed by SARS-CoV-2 peptide pool stimulation (*p* < 0.05 versus SARS-CoV-2 peptide pool-stimulated group) (Figure 7).

## 3. Discussion

This study sought to explore the effect of Ang II stimulation followed by SARS-CoV-2 peptide pool stimulation on PBMCs from recovered COVID-19 patients. SARS-CoV-2-specific memory response was heightened in the presence of Ang II by increasing CD38+, CD69+, and CD127+ SARS-CoV-2-specific CD8+ and CD4+ T-cells (Figure 1 and Figure 3). CD38 constituted to a hallmark of severe COVID-19 [14]. CD38 and CD69 play role in signal transduction and proliferation during the activation of T-cell memory response [15]. Additionally, CD127 is critical for maintaining memory T-cells [16] and serves as a survival marker by activating IL-7 signaling [15], which induces the expression of anti-apoptotic proteins [17] and promotes the inflammatory function of Th1 cells [18]. Consistent with this notion, Ang II showed a functional polarization of mitogen-stimulated PBMCs toward inflammatory CD4+ Th1/Th17 and CD8+ Th17 phenotypes by increased production of proinflammatory IL-17, TNF-α, and IFN-γ (Figure 5) as key markers of pro-inflammatory cell phenotype. This study’s findings align with a previous assessment showing that the infusion of Ang II increased the Th17 production of proinflammatory IL-17, thereby maintaining vascular dysfunction [19]. Additionally, elevated levels of IL-17 were found to increase C-reactive protein [20], which was upregulated in COVID-19 severe cases and plays a crucial role during infection as an early defense innate immune system [21]. The increase of production of IL-17 in our model following SP stimulation confirmed the major role of Th17 cells in response to direct contact with the virus. Recent studies have also found a positive correlation between elevated Th17, IL-17 production, and COVID-19 severity [1,22,23].

Ang II plays a role in adaptive immunity during acute respiratory distress syndrome (ARDS) development by inducing proinflammatory IFN-γ, TNF-α, and IL-6 production [10,24]. Our results showed an increase of IFN-γ expressing CD4+ T-cells (Figure 4A and Appendix A) in the presence of Ang II stimulation. A previous study showed that Ang II infusion stimulates PBMCs by increasing the production of IFN-γ and TNF-α, which promotes the development of vascular dysfunction [25]. IFN-γ and TNF-α promote the production of other proinflammatory cytokines, leading to sustained inflammatory response, tissue damage, and immunothrombosis [24]. An uncontrolled production of proinflammatory mediators, such as IL-6, IL-1β, IFN-γ, and TNF-α [25], drives hyperinflammation during severe COVID-19 infection, leading to cardiovascular and respiratory complications, including ARDS, disseminated intravascular coagulation [26], and heart failure [27]. In the current study, ELISA showed that the stimulation of PBMCs with Ang II activated the inflammatory response through increased production of IL-1β and IL-6 without affecting the reduction of anti-inflammatory IL-10 production caused by SARS-CoV-2 peptide pool stimulation (Figure 6). Additionally, Ang II did not activate the anti-inflammatory CD4+ CD25+ regulatory T-cells (Appendix A), indicating the role of Ang II in triggering an inflammatory response and that compensatory systems physiologically stimulated during inflammation and following virus infection are not affected by Ang II.

In the event of COVID-19 infection, Th17 cells are involved in the cytokine storm by producing IL-17, which increases the production of inflammatory IL-1β, IL-6, and TNFα cytokines, leading to tissue damage, lung injury, and pulmonary edema [28,29]. Although the activation of Th1 and Th17 phenotypes is integral for viral clearance [30], excessive activation of these phenotypes, such as in the presence of Ang II (Figure 5), promotes chronic inflammation through the continuous production of proinflammatory mediators including IL-6 and IL-1β [31]. In consonance with these observations, a recent comparative study showed an overactivation of Th17 accompanied by the inhibition of regulatory T-cells in ICU patients with COVID-19 compared with in healthy controls, leading to tissue damage and hyperinflammation during COVID-19 [32]. 

CD154 (or CD40L), a member of the TNF family, is expressed on T-cells [33] and contributes to cytokine storm induction during COVID-19 infection [34]. The current study showed that CD154 and TNF-α expression levels were elevated in the presence of SARS-CoV-2 peptide pool stimulation or Ang II stimulation separately (Figure 2 and Figure 4B,C). By contrast, the stimulation of PBMCs with Ang II followed by SARS-CoV-2 peptide pool reduced CD154 and TNF-α expression in CD4+ and CD8+ T-cells (Figure 2 and Figure 4B,C), which could indicate the negative synergistic effect of both stimulators. CD154/CD40 signaling proves critical for host defense against pathogens and essential for humoral and cellular immunity. It activates T-cell-dependent B-cell proliferation, antibody synthesis, and proinflammatory cytokine production [34,35,36]. This study’s observation demonstrated a consistency with a recent assessment of respiratory poxvirus infection, which showed that the deficiency of TNF-α can worsen lung pathology, as it helps regulate inflammation [37]. The exact mechanism and implication of CD154 and TNF-α reduction require more investigation, as this study was limited by a small sample size.

The phosphokinase array analysis in pre-stimulated PBMCs with Ang II followed by SARS-CoV-2 peptide pool stimulation showed a significant increase in the phosphorylation of several proteins (Figure 7 and Appendix A) known to be implicated in inflammatory responses [38,39], cardiovascular diseases [40], and viral infection [41]. 

It was suggested that RASs contributed to cardiac hypertrophy and heart failure by CREB-mediated Ang II type 1 receptor (AT1R) activation [42]. The upregulation of AT1R expression during hypertension was shown to be mediated by CREB, which was activated by Ang II-dependent P38 MAPK [43] and ERK signaling [42]. Exogenous Ang II stimulation significantly increased the phosphorylation of CREB, ERK1/2, and P38α (Figure 7). Ang II-dependent P38 MAPK activation contributes to endothelial dysfunction in cardiovascular diseases and end-organ damage during hypertension [44]. ERK1/2 and p38 MAPK are protein kinases that play a role in T-cell activation and proliferation via CREB phosphorylation which is either protective or pathogenic depending on the type of antigen [38,45].

Furthermore, Hsp27 serves as a prognostic biomarker for different ailments, such as heart failure [46] and coronary artery disease [47,48]. The phosphokinase array showed that Ang II stimulation significantly increased Hsp27 and PDGFR β activity (Figure 7). The activation of PDGFR β contributes to Ang II-induced vascular hypertrophy during hypertension [49]. Rajaiya et al. revealed that during viral infection, Hsp27 proves essential for p38 MAPK-mediated proinflammatory cytokine expression in host cells [50,51], and the activation of PDGFR β signaling facilitates viral entry during influenza infection [41], which may also apply during COVID-19 infections in the presence of comorbidities related to Ang II elevation.

Besides all the above-mentioned functional effects of the phosphorylated proteins, this study found that eNOS and Gsk-3β were significantly activated (Figure 7) in response to the combinational stimulation of PBMCs with Ang II followed by SARS-CoV-2 peptide pool activation compared to stimulation with a SARS-CoV-2 peptide pool alone (*p* < 0.05). The activation of eNOS, as shown in this study, might contribute to bradykinin-induced pulmonary edema during COVID-19 infection [52]. Bradykinin is an essential mediator of Ang II type 2 receptor-dependent eNOS activation [53] and contributes to tissue damage during inflammation [54]. In a recent study, NOS activity elevated in COVID-19 and related to the disease severity [22]. Moreover, Gsk-3 promotes inflammation and oxidative stress, which are integral events in ARDS development during COVID-19 infection. The activation of Gsk-3 induces the production of proinflammatory IL-1β, IL-6, TNF-α, and IFN-γ-supporting disease progression [55]. 

Collectively, all these observations indicated that previous stimulation with Ang II or the presence of RAS dysregulation such as in cardiovascular diseases, which is correlated with COVID-19 severity and morbidity, causes an exaggeration of inflammatory response against SARS-CoV-2 infection without affecting the anti-inflammatory response. There are limitations in the study. The sample size was small due to the vaccination program, and there was insufficient information about the medical history of each enrolled patient, potentially resulting in confounding bias. Moreover, the use of in vitro experiments might not reflect what occurs in vivo, so that all these findings require further investigation. In the future, more studies are needed to clarify the effect of antecedent RAS dysregulation on COVID-19 infection. Such an approach might improve clinical practice and management and provide improved treatment for patients.

The present study utilized Ang II-stimulated PBMCs from recovered COVID-19 patients to investigate the effect of SARS-CoV-2 spike glycoprotein stimulation in the presence of exogenous Ang II. This approach mimicked cases of COVID-19 patients with previous RAS dysregulation which mostly appears in hypertensive patients and people with cardiovascular history. However, other approaches utilizing PBMCs from COVID-19 patients with cardiovascular diseases or hypertension could contribute to interpreting the symptoms and complications that develop in COVID-19 patients with cardiovascular diseases and might explain the effect of RAS inhibitors administration during COVID-19 infection

## 4. Materials and Methods

### 4.1. Human Samples

All blood donors were recruited based on their clinical history of SARS-CoV-2 infection. Recovered COVID-19 patients (n = 18) were enrolled in this study after confirming their recovery by having negative polymerase chain reaction tests. Patients were enrolled from King Abdulaziz Medical City, Riyadh, Kingdom of Saudi Arabia. All the participants were between 32 and 50 years old with mild to moderate symptoms, without comorbidity, and no one was hospitalized or under medications. Ten of the participants were female, and eight were males. All the participants were not vaccinated. 

### 4.2. Peripheral Blood Mononuclear Cells (PBMCs) Isolation

For all samples, whole blood was collected in acid citrate dextrose tubes before being centrifuged at 800× *g* for 20 min to separate the cellular fraction and plasma. Next, the plasma was carefully removed from the cell pellet and stored at −80 °C. Density-gradient sedimentation using Ficoll-Paque was performed to isolated PBMCs. The isolated PBMCs were cryopreserved in cell recovery media containing 10% dimethyl sulfoxide (DMSO) (Gibco, Waltham, MA, USA), supplemented with 90% heat-inactivated fetal bovine serum (FBS) and stored in liquid nitrogen until their use in the assays.

### 4.3. T-Cell Stimulation and Treatment

Cryopreserved PBMCs from recovered COVID-19 patients were thawed and washed in RPMI-1460 media (Gibco, Waltham, MA, USA) supplemented with 10% FBS and 1% pen-strep (Gibco, Waltham, MA, USA) and cultured for 4 h at 37 °C. The next step saw each patient’s PBMCs divided into four groups: (1) the unstimulated control group: treated with 0.4% DMSO (Gibco, Waltham, MA, USA) for 24 h corresponding to the DMSO concentration used in the other groups; (2) the SARS-CoV-2 peptide pool group: stimulated with 1 μg/mL PepMix^TM^ SARS-CoV-2 spike glycoprotein (JPT, PM-WCPV-S-1) for 24 h; (3) the SARS-CoV-2 peptide pool and angiotensin II group: stimulated with 100 nM Ang II (Calbiochem, San Diego, CA, USA) for 48 h and then 1 μg/mL SARS-CoV-2 peptide pool (JPT, PM-WCPV-S-1) added in the last 24 h; and (4) the angiotensin II group: stimulated with 100 nM Ang II (Calbiochem, USA) for 48 h. These groups’ treatments were used for further assays (Figure 8). The PepMix^TM^ SARS-CoV-2 spike glycoprotein contained two peptide pools. The first peptide pool consisted of 158 15-mers covering 1 to 643 amino acid residues with eleven amino acids overlapping, whereas the second peptide pool consisted of 157 15-mers covering 633 to 1271 amino acid residues with 11 amino acids overlapping. Both peptide pools were abbreviated as the “SARS-CoV-2 peptide pool”.

### 4.4. Activation-Induced Cell Marker Assay (AIM)

Cell surface markers activation was assessed by culturing PBMCs in 96-wells plates and stimulating them with 0.4% DMSO, a 1 µg/mL SARS-CoV-2 peptide pool, 100 nM Ang II, or both a 1 µg/mL SARS-CoV-2 peptide pool and 100 nM Ang II for 24 and 48 h, separately. Supernatants were collected post-stimulation and stored at −20 °C for ELISA. The pellets were washed and stained with the cell surface markers antibodies anti-CD4-PerCP5.5 (clone RM4-5; Invitrogen, Waltham, MA, USA), anti-CD8a-APC-eFluor 780 (clone RPA-T8; Invitrogen, Waltham, MA, USA), anti-CD25-PECY7 (Beckman Coulter, Brea, CA, USA), anti-CD38-E flour 450 (clone HIT2; eBioscience, Waltham, MA, USA), anti-CD69-FITC (clone H1.2F3; Invitrogen, Waltham, MA, USA), anti-CD127-PE (clone eBioRDR5; Invitrogen), anti-CD154-APC (clone 24-31; BioLegend, San Diego, CA, USA), and anti-CD8-FITC (Beckman Coulter, Brea, CA, USA) for 1 h on ice, before the cells were washed and acquired using a BD FACS Canto II Flow Cytometer (BD Biosciences, San Jose, CA, USA) as previously described [56]. The BD FACSDiva software was used to analyze the results. The gating strategy (Appendix A) was performed by BD LSRFortessa.

### 4.5. Intracellular Cytokine Staining

PBMCs were cultured in a 96-well plate and stimulated, as previously mentioned in the T-cells stimulation and treatment section, with an extra step involving the addition of 1 μL/200μL Monensin GolgiPlug (eBioscience, Waltham, MA, USA) after 6 h of SARS-CoV-2 peptide pool stimulation. During post-incubation, cells were washed, and the cell surface was stained with anti-CD4-PerCP5.5 (clone RM4-5; Invitrogen) and anti-CD8a-APC-eFluor 780 (clone RPA-T8; Invitrogen, Waltham, MA, USA) for 1 h on ice before being fixed with 1% paraformaldehyde and kept overnight at 4 °C. After permeabilization with 0.1% Tween 20 (Bio rad, Hercules, CA, USA) at room temperature for 20 min, the cells were intracellularly stained with TNF-α-PE (Beckman Coulter, Brea, CA, USA), IFN-γ Monoclonal Ab CC302 (Invitrogen, Waltham, MA, USA), and IFN-γ-Alexa Fluor 488 goat anti-mouse IgG1 (Invitrogen, Waltham, MA, USA) for 1 h on ice. Thereafter, the cells were washed and acquired using a BD FACS Canto II Flow Cytometer (BD Biosciences, San Jose, CA, USA). BD FACSDiva software was used to analyze the results. The gating strategy (Appendix A) was performed by BD LSRFortessa.

### 4.6. Human Phospho-Kinase Array

PBMCs were treated with a 1 µg/mL SARS-CoV-2 peptide pool, 100 nM Ang II, or a combination of a 1 µg/mL SARS-CoV-2 peptide pool and 100 nM Ang II for 24 and 48 h, separately. An equal amount of cell lysate (30 μg) from each group was used for the human phospho-kinase array kit (ARY003C; R&D Systems) to detect the relative protein phosphorylation levels according to the manufacturer’s instruction [57]. The Image Lab software was used to analyze signal intensity. The average spot signal value of the negative control was subtracted from the average spot signal value of each protein on the array.

### 4.7. Mitogenic Stimulation

PBMCs were cultured in 96-well plates and stimulated with 0.4% DMSO, a 1 µg/mL SARS-CoV-2 peptide pool, 100 nM Ang II, or a combination of a 1 µg/mL SARS-CoV-2 peptide pool and 100 nM Ang II for 24 and 48 h separately. Thereafter, the PBMCs were washed and restimulated with 50 ng/mL PMA, and 1 µg/mL Ionomycin, then 1 μL/200μL Monensin GolgiPlug (eBioscience, Waltham, MA, USA) was added for 6 h. During post-incubation, the cells were washed, and their surfaces were stained with anti-CD4-PerCP5.5 (clone RM4-5; Invitrogen, Waltham, MA, USA) and anti-CD8a-APC-eFluor 780 (clone RPA-T8; Invitrogen, Waltham, MA, USA) for 1 h on ice, then fixed with 1% paraformaldehyde and and incubated overnight at 4 °C. After permeabilization with 0.1% Tween 20 at room temperature for 20 min, the cells were intracellularly stained with TNF-α-PE (Beckman Coulter, Brea, CA, USA), IFN-γ Monoclonal Ab CC302 (Invitrogen, Waltham, MA, USA), IFN-γ-Alexa Fluor 488 goat anti-mouse IgG1 (Invitrogen), and 1L-17a-Alexa fluor 700 (Beckman Coulter, Brea, CA, USA) for 1 h on ice. Then, the cells were washed and acquired using a BD FACS Canto II Flow Cytometer (BD Biosciences, San Jose, CA, USA) as previously described [56]. The BD FACSDiva software was used to analyze the results. The gating strategy (Appendix A) was performed by BD LSRFortessa.

### 4.8. ELISA Assay

The ELISA assay took place using supernatants collected during the cell AIM assay. IL-6, IL-10, and IL-1β ELISA Kits from Solarbio Life Sciences (SEKH-0013, SEKH-0018, and SEKH-0002) were applied as directed in the user manual. In briefly, 100 µL of standard samples were added to each antibody-coated well and incubated at 37 ℃ for 90 min. Then, 100 µL of the biotin-conjugated detection antibody was added to each well and incubated at 37 ℃ for 60 min. Thereafter, 100 µL of streptavidin-HRP were added to each well and incubated at 37 ℃ for 30 min. Then, 100 µL of substrate solution (TMB) were added to each well and incubated at 37 ℃ for 15 min with protection from light. Finally, 50 µL of stop solutions were added to each well, and the plate was read at 450 nm within 30 min. Standard curves were constructed, and trendline equations were used to determine the IL-10, IL-6, and IL-1β concentrations. 

### 4.9. Statistical Analysis

A Student’s *t*-test was used to assess the statistical significance between two groups, and a *p*-value of less than 0.05 was considered statistically significant. Data were performed in triplicate and presented as mean ± SEM.

## 5. Conclusions

Our results implied the presence of RAS dysregulation, which was previously considered the primary contributor in cardiovascular pathology, before COVID-19 infection could worsen COVID-19 symptoms. Elevated Ang II levels appeared to aggravate the inflammatory response during COVID-19 infection, leading to potent cytokine storms and organ damage related to COVID-19 severity and mortality. Targeting RASs might be a therapeutic strategy for critically ill cases of COVID-19. 

## Figures and Tables

**Figure 1 ijms-23-08669-f001:**
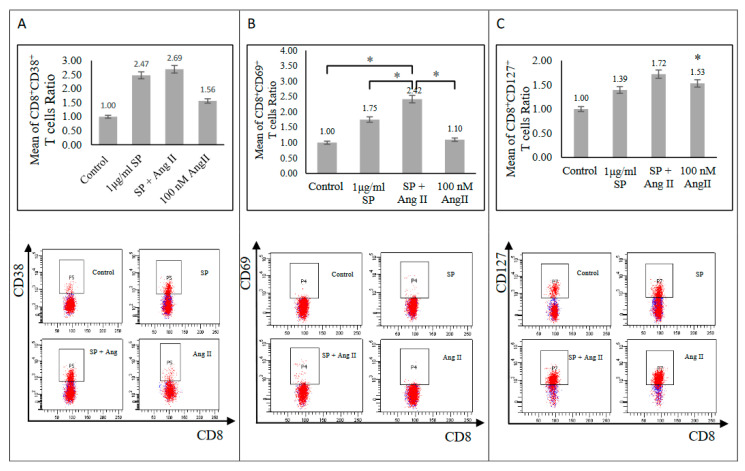
Activated CD8+ T-cells surface markers of recovered COVID-19 patients. (**A**) Upper: CD8+ CD38+ T-cells (n = 8). Lower: FACS representative plot. (**B**) Upper: CD8+ CD69+ T-cells (n = 9). Lower: FACS representative plot. (**C**) Upper: CD8+ CD127+ T-cells (n = 10). Lower: FACS representative plot. The ratio of CD8+ T-cells responses was measured as the mean of the percentage of the cell surface marker-positive T-cells in each group to the percentage of the cell surface marker-positive T-cells in the control group after the stimulation of PBMCs with a SARS-CoV-2 peptide pool (SP), angiotensin II (Ang II), or both of a SP and Ang II. *, *p* < 0.05.

**Figure 2 ijms-23-08669-f002:**
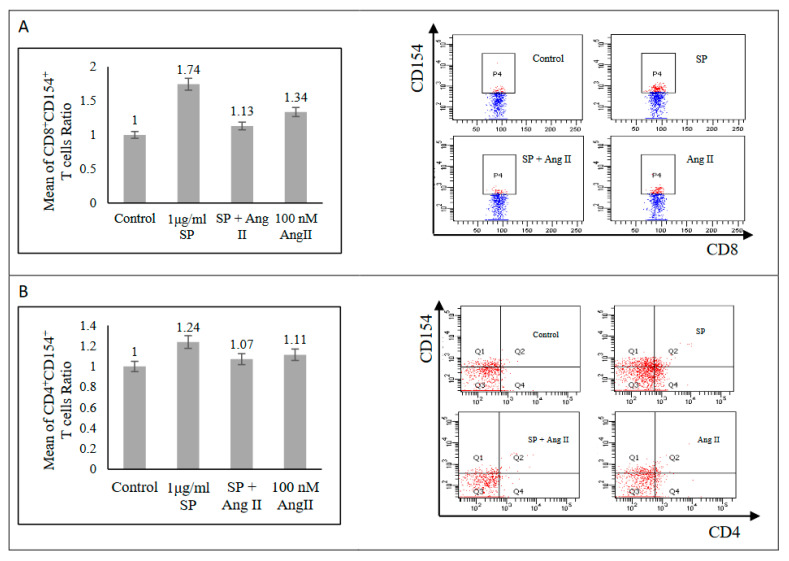
Activated T-cells surface markers of recovered COVID-19 patients. (**A**) Left: CD8+ CD154+ T-cells (n = 4). Right: FACS representative plot. (**B**) Left: CD4+ CD154+ T-cells (n = 4). Right: FACS representative plot. The ratio of T-cells responses was measured as the mean of the percentage of the cell surface marker-positive T-cells in each group to the percentage of the cell surface marker-positive T-cells in the control group after the stimulation of PBMCs with a SARS-CoV-2 peptide pool (SP), angiotensin II (Ang II), or both of a SP and Ang II.

**Figure 3 ijms-23-08669-f003:**
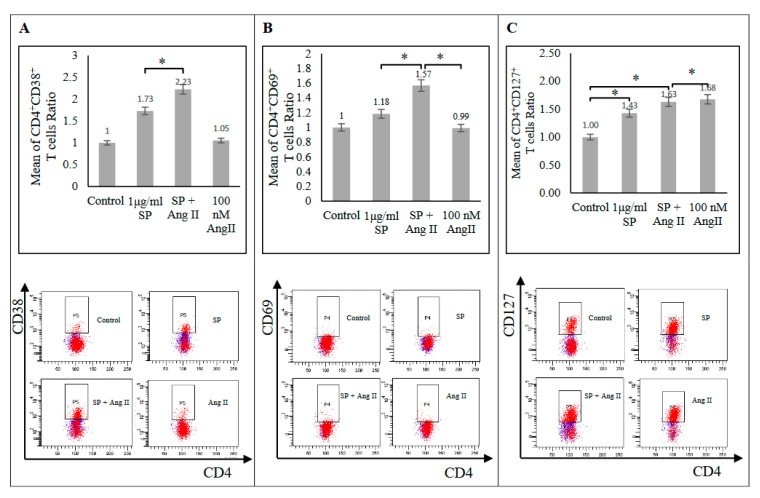
Activated CD4+ T-cells surface markers of recovered COVID-19 patients. (**A**) Upper: CD4+ CD38+ T-cells (n = 6). Lower: FACS representative plot. (**B**) Upper: CD4+ CD69+ T-cells (n = 7). Lower: FACS representative plot. (**C**) Upper: CD4+ CD127+ T-cells (n = 8). Lower: FACS representative plot. The ratio of CD4+ T-cells responses was measured as the mean of the percentage of the cell surface marker-positive T-cells in each group to the percentage of the cell surface marker-positive T-cells in the control group after the stimulation of PBMCs with a SARS-CoV-2 peptide pool (SP), angiotensin II (Ang II), or both of a SP and Ang II. *, *p* < 0.05.

**Figure 4 ijms-23-08669-f004:**
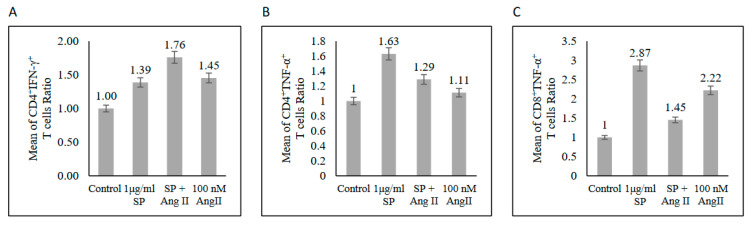
Ratios of CD4+ T-cells producing IFN-γ (**A**), TNF-α (**B**), and CD8+ T-cells (**C**) producing TNF-α in response to the SARS-CoV-2 peptide pool (SP), angiotensin II (Ang II), or both of SP and Ang II stimulation in PBMCs from recovered COVID-19 patients (n = 5). The mean of the T-cells ratios was measured as the percentage of T-cells producing cytokine in each group to the percentage of T-cells producing cytokine in the control group.

**Figure 5 ijms-23-08669-f005:**
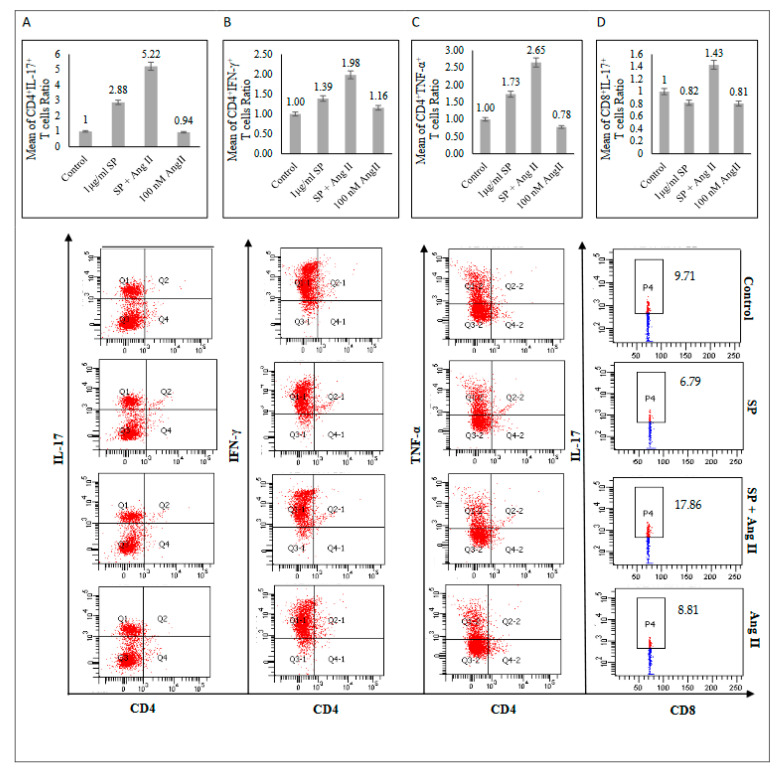
Functional profile of T-cells from recovered COVID-19 patients (n = 5) after mitogenic stimulation. (**A**) Upper: CD4+ IL-17+. Lower: FACS representative plot. (**B**) Upper: CD4+ IFN-γ+. Lower: FACS representative plot. (**C**) Upper: CD4+ TNF-α+. Lower: FACS representative plot. (**D**) Upper: CD8+ IL-17+. Lower: FACS representative plot. The mean of the T-cells ratios was measured as the percentage of T-cells producing cytokine in each group to the percentage of T-cells producing cytokine in the control group.

**Figure 6 ijms-23-08669-f006:**
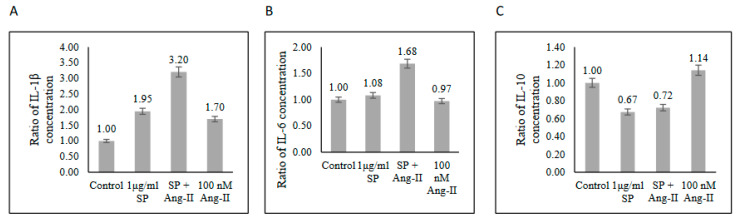
Cytokine levels in the supernatant of PBMCs from recovered COVID-19 patients (n = 3) after stimulation with a SARS-CoV-2 peptide pool (SP), angiotensin II (Ang II), or both of SP and Ang II. (**A**) IL-1β. (**B**) IL-6. (**C**) IL-10. The ratio of cytokines concentrations was measured as the concentration of cytokine in each group to the concentration of cytokine in the control group.

**Figure 7 ijms-23-08669-f007:**
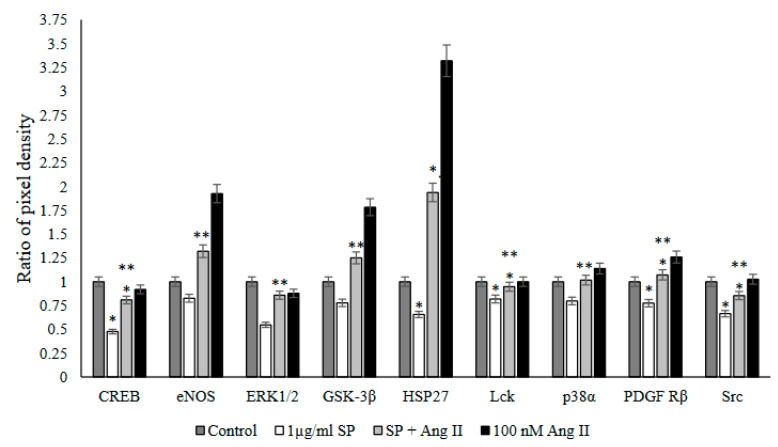
Phospho-kinase array analysis of PBMCs from recovered COVID-19 patients (n = 13) after stimulation of PBMCs with a SARS-CoV-2 peptide pool (SP), angiotensin II (Ang II), or both of a SP and Ang II. The ratio of pixel density was measured as the mean of the pixel density of each group to the mean of the pixel density of the control group. Out of the 21 tested proteins, only the proteins which showed significant changes in phosphorylation by both of the SARS-CoV-2 peptide pool (SP) and angiotensin II (Ang II) stimulation are shown. *, *p* < 0.05 vs. control group; **, *p* < 0.05 vs. 1 µg/mL SARS-CoV-2 peptide pool group.

**Figure 8 ijms-23-08669-f008:**
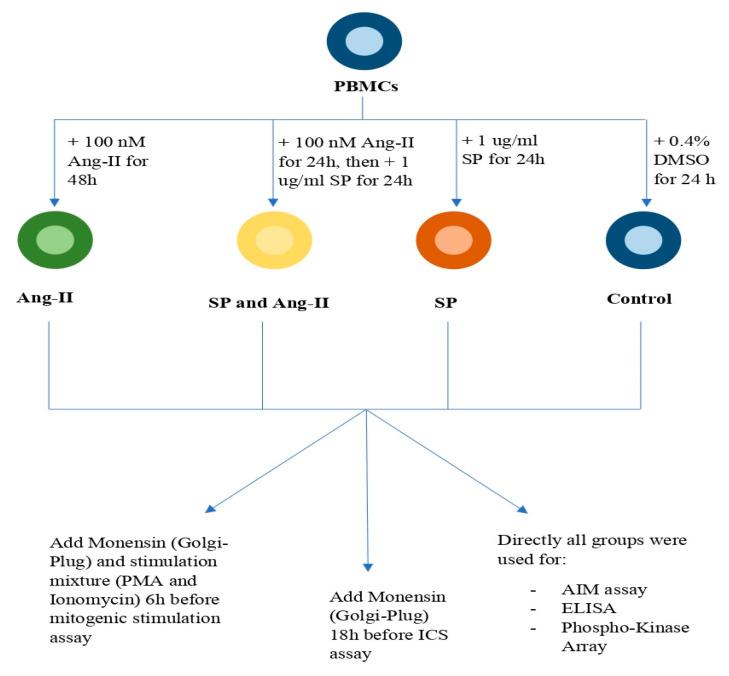
Protocol summary. Activation-induced cell marker assay (AIM); angiotensin II (Ang II); dimethyl sulfoxide (DMSO); enzyme-linked immunosorbent assay (ELISA); intracellular cytokine staining (ICS); peripheral blood mononuclear cells (PBMCs); phorbol 12-myristate 13-acetate (PMA); SARS-CoV-2 peptide pool (SP).

## Data Availability

Not applicable.

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
