# Peer review of "Angiotensin II Exaggerates SARS-CoV-2 Specific T-Cell Response in Convalescent Individuals following COVID-19"

_ijms, 2022, doi:10.3390/ijms23158669_

Round 1
Reviewer 1 Report
No further comments.
Author Response
Open Review
(x) I would not like to sign my review report
( ) I would like to sign my review report
English language and style
( ) Extensive editing of English language and style required
( ) Moderate English changes required
(x) English language and style are fine/minor spell check required
( ) I don't feel qualified to judge about the English language and style
|
Yes |
Can be improved |
Must be improved |
Not applicable |
|
|
Does the introduction provide sufficient background and include all relevant references? |
(x) |
( ) |
( ) |
( ) |
|
Are all the cited references relevant to the research? |
(x) |
( ) |
( ) |
( ) |
|
Is the research design appropriate? |
(x) |
( ) |
( ) |
( ) |
|
Are the methods adequately described? |
(x) |
( ) |
( ) |
( ) |
|
Are the results clearly presented? |
(x) |
( ) |
( ) |
( ) |
|
Are the conclusions supported by the results? |
(x) |
( ) |
( ) |
( ) |
Comments and Suggestions for Authors
No further comments.
- Response: Thank you for taking the time and effort necessary to review our manuscript.

Reviewer 2 Report
This study aims to investigate the effect of exogenous Ang-II on SARS-CoV-2 specific T-cells response in COVID-19 recovered patients.
Based on their results, the authors concluded that Ang-II increases SARS-CoV-2 specific T-cells response, aggravating the inflammatory response and changing the immune response.
The article is interesting. I have a few suggestions before publication:
- I suggest improving the discussion section. The authors remark on their results and this choice is redundant with the results sections. I suggest summarizing better their results and improving the comparison with international data.
Minor points:
- the short version should be preceded by the extended version: please, check all abbreviations starting from the Abstract section
Author Response
Reviewer 2
Open Review
( ) I would not like to sign my review report
(x) I would like to sign my review report
English language and style
( ) Extensive editing of English language and style required
( ) Moderate English changes required
(x) English language and style are fine/minor spell check required
( ) I don't feel qualified to judge about the English language and style
|
Yes |
Can be improved |
Must be improved |
Not applicable |
|
|
Does the introduction provide sufficient background and include all relevant references? |
( ) |
(x) |
( ) |
( ) |
|
Are all the cited references relevant to the research? |
( ) |
(x) |
( ) |
( ) |
|
Is the research design appropriate? |
( ) |
(x) |
( ) |
( ) |
|
Are the methods adequately described? |
( ) |
(x) |
( ) |
( ) |
|
Are the results clearly presented? |
( ) |
(x) |
( ) |
( ) |
|
Are the conclusions supported by the results? |
( ) |
(x) |
( ) |
( ) |
Comments and Suggestions for Authors
This study aims to investigate the effect of exogenous Ang-II on SARS-CoV-2 specific T-cells response in COVID-19 recovered patients.
Based on their results, the authors concluded that Ang-II increases SARS-CoV-2 specific T-cells response, aggravating the inflammatory response and changing the immune response.
The article is interesting. I have a few suggestions before publication:
- I suggest improving the discussion section. The authors remark on their results and this choice is redundant with the results sections. I suggest summarizing better their results and improving the comparison with international data.
- Response: Thank you for your comments, it has been improved according to your suggestion.
Minor points:
- the short version should be preceded by the extended version: please, check all abbreviations starting from the Abstract section
- Response: It has been checked and modified
